# Human iPSC- and Primary-Retinal Pigment Epithelial Cells for Modeling Age-Related Macular Degeneration

**DOI:** 10.3390/antiox11040605

**Published:** 2022-03-22

**Authors:** Cody R. Fisher, Mara C. Ebeling, Zhaohui Geng, Rebecca J. Kapphahn, Heidi Roehrich, Sandra R. Montezuma, James R. Dutton, Deborah A. Ferrington

**Affiliations:** 1Department of Ophthalmology and Visual Neurosciences, University of Minnesota, Minneapolis, MN 55455, USA; fishe765@umn.edu (C.R.F.); ebeli017@umn.edu (M.C.E.); kapph001@umn.edu (R.J.K.); smontezu@umn.edu (S.R.M.); 2Graduate Program in Biochemistry, Molecular Biology, and Biophysics, College of Biological Sciences, University of Minnesota, Minneapolis, MN 55455, USA; 3Stem Cell Institute, University of Minnesota, Minneapolis, MN 55455, USA; gengx027@umn.edu; 4Department of Genetics, Cell Biology, and Development, University of Minnesota, Minneapolis, MN 55455, USA; 5Histology Core for Vision Research Department of Ophthalmology and Visual Neurosciences, University of Minnesota, Minneapolis, MN 55455, USA; rohri002@umn.edu

**Keywords:** retinal pigment epithelium, age-related macular degeneration, mitochondria, oxidative stress

## Abstract

Primary cultures of retinal pigment epithelium (RPE) from human adult donors (haRPE) and induced pluripotent stem cell derived-RPE (iPSC-RPE) are valuable model systems for gaining mechanistic insight and for testing potential therapies for age-related macular degeneration (AMD). This study evaluated the treatment response of haRPE and iPSC-RPE to oxidative stress and potential therapeutics addressing mitochondrial defects. haRPE and iSPC-RPE were derived from donors with or without AMD. Mitochondrial function was measured after treatment with menadione, AICAR, or trehalose and the response to treatment was compared between cell models and by disease status. In a subset of samples, haRPE and iPSC-RPE were generated from the same human donor to make a side-by-side comparison of the two cell models’ response to treatment. Disease-specific responses to all three treatments was observed in the haRPE. In contrast, iPSC-RPE had a similar response to all treatments irrespective of disease status. Analysis of haRPE and iPSC-RPE generated from the same human donor showed a similar response for donors without AMD, but there were significant differences in treatment response between cell models generated from AMD donors. These results support the use of iPSC-RPE and haRPE when investigating AMD mechanisms and new therapeutics but indicates that attention to experimental conditions is required.

## 1. Introduction

Age-related macular degeneration (AMD) is the leading cause of blindness in the elderly, with an estimated 288 million cases of AMD worldwide by 2040 [1]. There are two clinically distinct forms of the disease, wet AMD and dry AMD. Wet AMD, resulting from abnormal growth of blood vessels into the retina, has several effective treatments available to prevent vision loss. The dry form of AMD is a progressive disease that culminates in central vision impairment due to the death of the retinal pigment epithelium (RPE) and subsequent loss of the light-sensing photoreceptors. The RPE performs multiple functions that support retinal function, including the secretion of growth factors, transport of nutrients and oxygen from the outer retina blood supply, and renewal of photoreceptor outer segments via phagocytosis [2]. Currently, there are no effective therapies to treat dry AMD, which affects a majority of AMD patients. Discovering potential targets for therapy and testing promising treatment candidates requires practical model systems that authentically replicate disease phenotypes.

Chronic oxidative stress is a well-recognized factor in AMD pathobiology [3]. A major source of RPE oxidative stress is from reactive oxygen species (ROS) generated in the mitochondria. Mitochondria-derived ROS can damage DNA, proteins, and lipids in the cytosol and within the mitochondria. Strong experimental evidence from studies in human donor retinas supports the hypothesis that ROS-driven mitochondrial dysfunction plays a central role in AMD pathology [4].

Due to their role in retinal health and their eventual death with AMD progression, the RPE is a key target for AMD treatments. There are a number of in vitro human RPE cell models, including human adult primary RPE (haRPE) and RPE derived from induced pluripotent stem cells (iPSC-RPE). haRPE cultures from deceased human donors have provided valuable insight into AMD phenotypes and disease mechanisms and have been used to investigate the efficacy of potential therapeutics [5,6,7,8]. However, this system has challenges, including the poor availability of donor tissue and limited ability to expand cell numbers in culture. iPSC-RPE are a well-characterized alternative model system with several important advantages, including the ability to produce large numbers of cells and their generation from multiple somatic cells sources (i.e., blood, conjunctiva, skin), which makes sampling from living patients possible [9,10,11,12]. Previous studies demonstrated metabolic dysfunction and altered gene expression in iPSC-RPE from AMD donors [9,13,14,15,16], demonstrating the feasibility of using iPSC-RPE to investigate AMD disease mechanisms. Another benefit of iPSC-RPE is their use in cell therapies designed to replace the RPE layer, which is currently in pre-clinical trials [17]. Finally, the nearly limitless supply of iPSC-RPE provide the opportunity for both large scale drug screening platforms as well as patient-specific testing [12,14,15,18,19,20,21].

With each experimental system, it is essential to understand how closely the model system replicates the target cell in vivo. Our group and others have shown that haRPE and iPSC-RPE cultures exhibit many cardinal features of native RPE, including the formation of a pigmented epithelial layer with tight junctions, and the expression of RPE signature genes and proteins [5,6,9,10,14]. Additionally, both cell models phagocytose outer segments and attain correct apical and basal polarity, as demonstrated by specific protein localization and directional secretion of growth factors [5,8,16,18]. While both cell systems exhibit morphological and functional similarities, in-depth analysis is required to reveal potential cryptic differences relevant to modeling AMD.

In prior investigations using haRPE, we observed reduced mitochondrial function, increased resistance to oxidative stress, and a greater response to mitochondrial-targeted drugs in RPE cultured from donors with AMD [5,7]. Based on these AMD-associated differences, we chose three compounds to test whether iPSC-RPE can replicate the response of haRPE. We compared the mitochondrial function at baseline and after menadione treatment, which causes mitochondrial inhibition and oxidative stress by generating reactive oxygen species through redox cycling [22]. We also investigated the response of haRPE and iPSC-RPEs to AICAR (5-aminoimidazole-4-carboxamide ribonucleotide) and trehalose. AICAR was used to stimulate mitochondrial biogenesis, while trehalose was used to increase cellular autophagy. Data were analyzed as an aggregate of all donors, and for a subset of genetically identical haRPE and iPSC-RPE generated from the same donor. To the best of our knowledge, this is the first report of a side-by-side comparison of these two cell model systems generated from the same human source.

In the aggregate data, disease-specific differences were observed at baseline and in response to all three treatments in the haRPE. However, irrespective of disease status, iPSC-RPE had a similar response to all three treatments and matched the treatment response of haRPE from donors without AMD. Data from haRPE and iPSC-RPE generated from the same human donor showed a similar response to treatment when generated from donors without AMD. Similar to the aggregate data, iPSC-RPE and haRPE generated from donors with AMD had a significantly different response to treatments.

## 2. Materials and Methods

### 2.1. Cell Culture

Donor eyes were obtained with the informed consent of the donor or donor’s family for use in medical research in accordance with the Declaration of Helsinki. The Minnesota Grading System (MGS) was used to classify donor eyes into No AMD (MGS1) and AMD (MGS2 and MGS3) [23]. Evaluation for MGS stages was determined by a Board Certified Ophthalmologist (Dr. Sandra R. Montezuma). haRPE were generated and cultured as described previously [5,24]. The derivation of iPSC-RPE lines from primary human conjunctival cells, differentiation of iPSC to RPE, and expansion of iPSC-RPE were described in our previous publication [10]. In brief, conjunctival epithelial cells of donor eyes are used to generate induced pluripotent stem cells and then differentiated into iPSC-RPE. For both haRPE and iPSC-RPE, each passage was ~30 days, and cells from passage 3 were used for characterization and the functional assays. Representative images showing the pigment, morphology, and expression of proteins through immunohistochemistry are shown in Appendix A.

Both cell models were cultured using the same conditions and culture media beginning at passage 3 and throughout each assay, with media changes twice a week. Confluent cell layers were used in the experiments, with seeding density and plate type provided in each experimental method. The demographics for donors used to generate haRPE and iPSC-RPE are provided in Appendix A. Analysis of the mitochondrial function for males and females found no gender-specific differences, therefore their data is combined in this manuscript (Appendix A).

For drug treatments, cells were cultured in media containing 1% FBS. The doses and timing of all treatments are indicated in their respective figures. The treatment conditions for the final experiments were 25 μM menadione for 24 h, 500 μM AICAR for 48 h, and 100 mM trehalose for 48 h.

### 2.2. Genotyping

Genomic DNA was extracted from graded donor retinal tissue as described previously [12]. Samples were genotyped for the Complement Factor H (CFH) variant Y402H (SNP; rs1061170) or ARMS2 variant A69S (SNP; rs10490924) using allele-specific primers designed for each SNP. CFH-Y402H-F: TGAGGGTTTCTTCTTGAAAATCA, CFH-Y402H-R: CCATTGGTAAAACAAGGTGACA, ARMS2-A69S-F: TCCTGGCTGAGTGAGATGG, ARMS2-A69S-R: GGCATGTAGCAGGTGCATT.

### 2.3. Real-Time PCR

Approximately 300,000 cells were plated and grown to confluence in 12-well plates. Cells were treated with varying concentrations (10, 25, 50 µM) of menadione for 24 h. RNA was collected, and cDNA was synthesized and quantified as previously described [16]. Real-time PCR was performed to analyze the gene expression of antioxidant genes (HO-1 and SOD2). Primer sequences can be found in Appendix A.

### 2.4. LysoTracker

haRPE and iPSC-RPE (40,000 cells per well) were plated into clear-bottom black-sided 96-well plates (Costar) and incubated for up to 48 h before staining and imaging. Cells were stained with 100 nM Lysotracker Red DND-99 (Fisher Scientific) and NucBlue Live Cell Stain (Invitrogen) and imaged as previously described [12].

### 2.5. Measuring Mitochondrial Function

Mitochondrial function in treated and untreated RPE was measured using the XFe96 Extracellular Flux Analyzer (Agilent Technologies, Santa Clara, CA, USA) and the Cell Mito Stress Test (CMST) assay. In XFe96-well plates, 40,000 cells were plated and incubated for up to 2 days before treatment and CMST assays. The CMST assay protocol was performed according to the manufacturer’s instructions (Agilent Technologies) and our previous publication [5,7,16]. Oxygen consumption rate (OCR) traces were used to calculate basal respiration (BR), ATP-linked respiration (ATP), spare respiratory capacity (SRC), and maximal respiration (MR). Data were normalized to cell count from 10× images taken after the CMST assay using a Cytation 1 (BioTek). The data processing used Wave software (Agilent Technologies).

### 2.6. Cell Death Assays

Sub-confluent (5000 cells/well) were plated into a ½ area 96-well clear-bottom, black-sided plates. After treatment, cell death was determined using CyQuant (Thermo Fisher) following the manufacturer’s protocol. Data were calculated relative to no treatment and lysis control wells. Cell death from confluent cells (40,000/well) was calculated from Seahorse plate cell counts after CMST assay.

### 2.7. Enzyme-Linked Immunosorbent Assay (ELISA)

haRPE and iPSC-RPE (40,000 cells per well) were plated in 6.5 mm transwell inserts and cultured for 5 weeks. Media was collected from apical and basal chambers and ELISA for vascular endothelial growth factor A (VEGF-A; eBioscience BMS277/2) and pigment epithelium-derived factor (PEDF; R&D Systems DY1177-05) were performed as described [12,16].

### 2.8. Western Blotting

haRPE and iPSC-RPE (~300,000 cells per well) were plated into 12 well-plates and whole cell lysates were collected using RIPA buffer after treatment (Sigma-Aldrich, St. Louis, MO, USA). Protein concentrations were determined with BCA assay (Thermo Scientific, Waltham, MA, USA) using albumin as the standard. 10 μg of protein was loaded for each sample and Western blots were performed as described [12]. Membranes were incubated overnight with primary antibodies (see Appendix A). Images of immune reactions were taken using a BioRad ChemiDoc XRS. Representative images are provided in Appendix A.

### 2.9. Statistics and Treatment Calculations

The data shown were calculated relative to no treatment controls for each donor (fold change relative to no treatment). The student’s *t*-test was used to compare treatment effects to no treatment controls as well as to compare haRPE and iPSC-RPE (Figures 1–4). A paired t-test was used to compare haRPE and iPSC-RPE generated from the same human donor (Figure 5). Statistical analysis was performed using Graphpad Prism 9. *p* ≤ 0.05 was considered statistically significant.

## 3. Results

### 3.1. Characterization of haRPE and iPSC-RPE under Basal Conditions

Donor demographics are provided in Appendix A. haRPE cultures were obtained from donors without AMD (No AMD; *n* = 12; aged 71 ± 7.5) and with AMD (AMD; *n* = 16; aged 75 ± 8.3). iPSC-RPE were generated from donors without AMD (No AMD; *n* = 7; aged 70 ± 11.4) and with AMD (AMD; *n* = 13; aged 75 ± 8.6). There was no significant difference between age ranges of haRPE and iPSC-RPE for No AMD (*p* = 0.82) and AMD (*p* > 0.99). Both cell models had a nearly balanced gender distribution, with the ratio of males and females 16:12 for haRPE and 9:11 for iPSC-RPE.

Prior to treatment, we first assessed haRPE and iPSC-RPE under basal conditions (Figure 1). To verify that both cell models produce growth factors with directed secretion that is consistent with RPE in vivo, we quantified content of VEGF-A and PEDF in the apical and basal chambers of transwell plates. As expected, both haRPE and iPSC-RPE had significantly greater VEGF-A and PEDF secreted in the basal and apical chambers, respectively (Figure 1A,C). While iPSC-RPE secreted more VEGF-A than the haRPE, the ratio of apical/basal VEGF-A was not significantly different between the two cell types (Figure 1B). When comparing the ratio of apical/basal PEDF, iPSC-RPE had ~2.5-fold increase relative to haRPE (Figure 1D).

The mitochondrial function in haRPE and iPSC-RPE was compared under basal conditions by measuring the oxygen consumption rate (OCR) using an XF Extracellular Flux Analyzer and the Cell Mito Stress Test (CMST). A comparison of mitochondrial function by disease status found that haRPE from AMD donors have lower mitochondrial function, whereas iPSC-RPE had no significant differences based on disease (Figure 1E,F). These results support previously published reports from our lab [5,16]. We next compared the two cell models, grouping them by disease status. In our sample of No AMD donors, maximal respiration (MR) and spare respiratory capacity (SRC) were significantly lower in iPSC-RPE compared to haRPE, while basal respiration (BR) and ATP-linked respiration (ATP) were similar (Figure 1G,H). In contrast, haRPE and iPSC-RPE from AMD donors exhibited similar mitochondrial function (Figure 1I,J). These data show that under basal conditions, the mitochondrial functional parameters of iPSC-RPE are most closely aligned with haRPE from AMD donors.

### 3.2. Response to Mitochondrial Oxidative Stress

Treatment with menadione was used to determine the cell’s response to mitochondrial oxidative stress. To establish the optimal dose, cells were treated with increasing concentrations of menadione for 24 h and then assayed for antioxidant gene expression, cell death, mitochondrial protein content, and mitochondrial function. The expression of the antioxidant genes HO-1 and SOD2, whose proteins are localized to the cytosol and mitochondria, respectively, was used to gauge the response to oxidative stress. Elevated expression of both HO-1 and SOD2 was observed in haRPE but was limited to only SOD2 in iPSC-RPE (Appendix A). Since cell density can affect the RPE response to oxidative stress [25,26], we assessed cell death under both non-confluent (5 k cells/well) and confluent (40 k cells/well) conditions. Under non-confluent conditions, haRPE exhibited a dose-dependent decrease in viability, with a significant 25% decrease at 20 µM menadione (Figure 2A). In contrast, non-confluent iPSC-RPE had significant cell death at all tested concentrations of menadione (Figure 2I). Under confluent conditions used in the CMST assay, cells showed greater resistance to treatment with no cell loss observed in either haRPE or iPSC-RPE (Figure 2C,K).

In measuring mitochondrial function, menadione caused a dose-dependent decrease in both haRPE and iPSC-RPE. haRPE had a significant decrease in BR, MR, SRC, and ATP at 50 µM menadione (Figure 2B). iPSC-RPE had significantly decreased MR and SRC with 25 µM menadione, and significant decreases in all parameters with 50 µM menadione (Figure 2K). Based on the cell count data (Figure 2C,K), the observed decrease in function is not linked to cell death. To investigate whether the loss in function was due to a loss of mitochondrial content, a panel of mitochondrial proteins were quantified following menadione treatment. Of these proteins, only COX II had a significant decrease in content with 25 µM menadione in haRPE (Figure 2B). The content of mitochondrial oxidative phosphorylation proteins remained unchanged in iPSC-RPE (Figure 2J). These results suggest the menadione-induced decrease in mitochondrial function was not due to the overall loss of mitochondria.

Based on these data, we chose 25 µM menadione, which caused an approximate 35% and 45% decrease in mitochondrial function in haRPE and iPSC-RPE, respectively. Our aggregate cohort of haRPE and iPSC-RPE were treated with 25 µM menadione for 24 h before measuring mitochondrial function. To understand how disease status effects treatment response, we grouped both haRPE and iPSC-RPE by the donor’s disease state. Following treatment, haRPE from No AMD donors had substantial decreases in every parameter, with significant decreases to MR and SRC (Figure 2E,F). In agreement with previously published data [5], haRPE from AMD donors were more resistant to stress, with only a significant decrease in ATP (Figure 2G,H). iPSC-RPE derived from either No AMD or AMD donors had a significant ~50% decrease in all calculated mitochondrial parameters after treatment with menadione (Figure 2M–P).

### 3.3. Response of RPE Cell Models to AICAR

AICAR, an analog of AMP, acts as a direct activator of adenosine-monophosphate-activated protein kinase (AMPK). Upon activation, AMPK phosphorylates a number of transcription factors that stimulate mitochondrial biogenesis and regulate metabolism [27]. Cells were treated for 1, 3, 6, 24, and 48 h with 500 µM AICAR, a dose selected based on our published work [12]. haRPE exhibited a sustained 2-fold increase in phospho-AMPK(Thr172)/Total AMPK at 3, 24, and 48 h of AICAR treatment (Figure 3A). iPSC-RPE had a significant 3-fold increase in AMPK activation at 1 and 3 h of AICAR but returned to baseline by 6 h (Figure 3G). Biogenesis was estimated by measuring mitochondrial proteins. haRPE showed a rapid increase in UQCRC2, COX II, and NDUFB8 content, with all proteins increased by 48 h (Figure 3B). Quantification of mitochondrial proteins in iPSC-RPE show a more limited response to AICAR with only a significant increase in COX IV content by 24 and 48 h (Figure 3H). These results indicate that AICAR treatment activated AMPK, stimulating mitochondrial biogenesis, and had a larger effect in haRPE.

We subsequently used 500 µM AICAR for 48 h to test its effect on mitochondrial function, grouping the aggregate cohort of haRPE and iPSC-RPE by the donor’s disease state. In haRPE and iPSC-RPE from No AMD donors, AICAR treatment significantly decreased BR and ATP (Figure 3C,D,I,J). In cells from AMD donors, AICAR significantly increased MR and SRC in haRPE, but had minimal effect in iPSC-RPE (Figure 3E,F).

### 3.4. Response of RPE Cell Models to Trehalose

Trehalose is a naturally occurring sugar that increases autophagy via AKT and the transcription factor TFEB, leading to the upregulation of genes associated with the Coordinated Lysosomal Expression and Regulation (CLEAR) network [28]. We utilized trehalose to remove damaged mitochondria through increased mitochondrial autophagy. haRPE and iPSC-RPE were treated with 100 mM trehalose for up to 48 h, in optimal conditions determined previously in our lab [12,28]. After trehalose treatment, haRPE and iPSC-RPE showed a substantial increase in Lysotracker™ staining, indicative of an expanded lysosomal compartment (Figure 4A,G). Consistent with this observation, there was a significant increase in lysosomal proteins LAMP1 and active Cathepsin D, as well as an increase in autophagy markers LC3-II/LC3-I and p62 (Figure 4B,H). When assessing trehalose’s effect on mitochondrial function, both haRPE and iPSC-RPE were generally unresponsive (Figure 4C,D,I–L), with a small but significant increase in MR and SRC in iPSC-RPE from No AMD donors.

### 3.5. Paired Comparison of RPE from the Same Donor

Our laboratory and others have shown that the genetic background of an individual can influence both the mitochondrial function and response to treatments [12,16,17,29,30]. To control for genetic differences, we generated haRPE and iPSC-RPE from the same human donor and performed side-by-side comparisons of the cell models. The demographics of the paired cell lines, including the genotype of the CFH and ARMS2 risk allele, are shown in Table 1.

To reduce technical variability, cells from the same donor were treated and assayed for mitochondrial function on the same plate. Data were analyzed by a paired *t*-test to determine if the response of haRPE and iPSC-RPE were significantly different. Bar graphs show the average response to each treatment with data points from individual donors superimposed over each bar, along with the results of the paired *t*-test (Figure 5A,B,D,E,G,H). We found no significant difference between haRPE and iPSC-RPE generated from donors without AMD. When comparing haRPE and iPSC-RPE generated from AMD donors, we observed significant differences in their response to menadione and AICAR (Figure 5A,D). Trehalose treatment showed no significant differences between haRPE and iPSC-RPE, likely due to the limited effect on mitochondrial function as measured by the CMST assay.

Data were also examined by individual pairs by averaging the absolute value of difference between haRPE and iPSC-RPE for each functional parameter. The summary shows considerable variation in donors, with the average difference being as low as 0.04 in some donors, and up to 0.68 in other donors depending on the treatment (Figure 5C,F,I). We found that paired haRPE and iPSC-RPE generated from donors without AMD were more similar (i.e., had a lower average difference) than those generated from donors with AMD for each of the three treatments. These data are consistent with findings from the aggregate data (Figure 1, Figure 2, Figure 3 and Figure 4).

## 4. Discussion

The purpose of this study was to identify shared and divergent characteristics of haRPE and iPSC-RPE. While a few studies have demonstrated that stem cell derived-RPE have proteomic and genomic similarities to primary RPE, our focus was on the mitochondria due to substantial evidence, from this study (Figure 1) and our prior publications, that link mitochondrial defects with AMD pathology [7,12,16,31,32,33,34]. Our analysis of the mitochondrial function and response to mitochondrial-targeted treatments revealed both similarities and differences between iPSC-RPE and haRPE. Similarities were observed in the treatment response of iPSC-RPE, irrespective of disease state, and haRPE from donors without AMD. Similarly, the results for haRPE and iPSC-RPE from donors without AMD were consistent for both the aggregate data (Figure 1, Figure 2, Figure 3 and Figure 4) and the comparison of haRPE and iPSC-RPE generated from the same donor (Figure 5), where there was a small magnitude of difference between the two cell models. In contrast, more differences were found when comparing iPSC-RPE to haRPE in cells generated from AMD donors.

In both the aggregate data (Figure 2 and Figure 3) as well as the paired cell lines from AMD donors (Figure 5), significant differences in the response to menadione and AICAR were observed. Menadione treatment decreased mitochondrial function in iPSC-RPE irrespective of disease status and in No AMD haRPE, but not in haRPE from AMD donors (Figure 2). The enhanced resistance of AMD haRPE against menadione-induced oxidative stress recapitulates our previous report of the enhanced cell survival of haRPE from AMD donors to hydrogen-peroxide induced oxidative stress [5]. Improved resistance to oxidative stress in unique donor populations using two different oxidants provides confidence that this is a reliable RPE biomarker for AMD. Differences were also found in the AICAR response. As expected, treatment with AICAR-activated AMPK in both haRPE and iPSC-RPE but the kinetics of activation differed (Figure 3). Rapid dephosphorylation of AMPK occurred in iPSC-RPE, while sustained AMPK phosphorylation was observed in haRPE. AICAR treatment increased the content of mitochondrial proteins in both haRPE and iPSC-RPE, although a larger effect was detected in haRPE and may be due to sustained AMPK activation. AICAR treatment led to decreases in two mitochondrial functional parameters in haRPE from No AMD donors, as well as in iPSC-RPE irrespective of disease state. However, AICAR had a positive effect on mitochondria in haRPE from AMD donors. This divergence in response was also evident in the large magnitude of difference in the two cell models generated from the same AMD donors (Figure 5) and supports previous findings from our lab that haRPE from donors with AMD are more responsive to drugs [7].

There are several potential explanations for the observed differences when comparing haRPE to iPSC-RPE. One idea is that the age of the cells may be an important factor. haRPE are isolated and cultured from the donor eyes of elderly individuals so the cell population has aged in vivo for at least 60 years. In contrast, reprogramming cells may remove the epigenetic signature of the adult somatic cell source, thereby reversing the “age” of iPSC-RPE. In our study, iPSC-RPE were in culture for only a few months and did not have the opportunity to age. Other studies support the idea of iPSC-RPE being in an immature state. One study found higher expression levels of progenitor genes like SOX2 and PAX6 in iPSC-RPE and lower expression of RPE-specific differentiation genes like TYR, RPE65, and BEST1, as compared with primary RPE cells [32]. Another study found that the stem cell-derived RPE proteome did not share signs of stress or changes associated with the degeneration that was observed in primary RPE [31].

The response and mitochondrial profile in haRPE from AMD donors were distinct not only from iPSC-RPE, but also the haRPE from donors without AMD. These results suggest distinct AMD-associated molecular changes are sustained in the cultured cells. It is plausible that haRPE generated from AMD donor eyes are the survivors of an in vivo diseased environment and have altered their proteome as a compensatory mechanism for survival. This “metabolic memory” has been observed in cells from diabetic patients and animal models, where the effect of the diseased environment is sustained in culture [1,35,36,37]. It has been suggested that the metabolic memory is due to epigenetic modifications and changes in gene expression that are retained in primary cultures.

In order to produce iPSC-RPE that more closely mimic the haRPE of AMD donors, in vitro manipulations may be needed. For example, it may be necessary to raise iPSC-RPE under conditions that more fully mimic the disease environment (i.e., chronic exposure to oxidants and cytokines). Other in vitro manipulations that replicate the diseased environment include growing iPSC-RPE on nitrited membranes [38], exposing iPSC-RPE to chronic oxidative stress by repeated treatments of peroxide [39], or culturing with media containing active complement [19]. These cell culture conditions helped to reveal AMD-associated differences in pathways, such as VEGF secretion, autophagy, and lipid deposition.

We found consistent results between groups of cells used in previous studies. For instance, the results within the haRPE group agree with results from previous reports of mitochondrial dysfunction, resistance to oxidative stress, and better responses to drug treatments in haRPE from AMD donors [5,6,7]. Results within the group of iPSC-RPE used in the current study also agree with our previous report in which similar mitochondrial function was detected in both No AMD and AMD iPSC-RPE using the CMST assay [16]. Nonetheless, other studies have suggested iPSC-RPE generated from AMD donors can display the disease phenotype including reduced metabolic function and altered gene expression of disease-related markers [9,13,14,15,40].

A number of factors could contribute to the different results and conclusions between studies. Most notably, the growth conditions of the iPSC-RPE vary from study to study. For example, cells grown on different types of matrixes can respond differently due to changes in the cellular environment [14,15,16,19,41]. Additionally, the components found in each media are often unique across publications, potentially influencing results. One study found the use of media containing active complement induces an AMD phenotype and atrophy in iPSC-RPE cells [19]. Lastly, the type of assay used to measure a specific outcome can influence the results. When measuring the mitochondrial function using the CMST assay, we found no significant difference in iPSC-RPE from donors with or without AMD in this study or previously with a separate cohort of donor lines [16]. However, when we used the Cell Energy Phenotype Test (Agilent) that stresses the mitochondria, a disease-related difference in mitochondrial function was revealed [16].

Our work and others have also found that the individual’s risk alleles can have a significant impact on RPE function. Two polymorphisms associated with AMD are complement factor H (CFH, rs1061170) and age-related maculopathy susceptibility 2 (ARMS2, rs10490924). Comparing the cells by genotype may reveal differences that are not observed when grouping the cells by donor disease state. Previous studies have reported that iPSC-RPE from donors with the high-risk allele for CFH have reduced mitochondrial function, increased inflammatory markers, and increased accumulation of lipid droplets [16,30,42]. Studies using iPSC-RPE with ARMS2/HTRA1 risk allele have reported decreased antioxidant defense, increased susceptibility to oxidative damage, and higher expression of inflammatory factors [14,29]. Due to the limitations of genotype diversity in the cell lines used, we were unable to conduct a genotypic comparison in this study.

While this study reveals that iPSC-RPE do not replicate the treatment response of diseased haRPE under current culture conditions, it does show the benefits of this valuable model system. We found that irrespective of disease state, iPSC-RPE match the response to treatment of haRPE from donors without AMD. This phenotype may benefit iPSC-RPE when transplanted into patient eyes, as they may better replenish and maintain the RPE layer upon transplant. Additionally, we observed an increased secretion of PEDF in the iPSC-RPE, which may prove beneficial in transplants as PEDF promotes development and survival of photoreceptors [43,44].

Other benefits of iPSC-RPE include their ability to produce large numbers of cells, and their generation from a number of cell types and living patients [12]. These benefits allow for patient-based and population studies as well as large-scale experiments to be performed on a single donor’s iPSC-RPE. While further comparisons, characterization, and standardization of methods is needed, iPSC-RPE remain a valuable cell model for studying AMD mechanisms and investigating potential therapeutics.

## 5. Conclusions

In conclusion, haRPE and iPSC-RPE are both important model systems for studying AMD. The current study highlights important differences and limitations of each cell culture system. We found that iPSC-RPE do not recapitulate the treatment response of haRPE from human donors with AMD but do replicate the response of haRPE from donors without AMD. The differences observed may be due to loss of epigenetic markers during iPSC-RPE differentiation or their growth in a non-diseased environment. The results of our study support the continued use of iPSC-RPE and haRPE when investigating AMD mechanisms and new therapeutics, but indicates that careful attention to the experimental conditions and donor genotype is required.

## Figures and Tables

**Figure 1 antioxidants-11-00605-f001:**
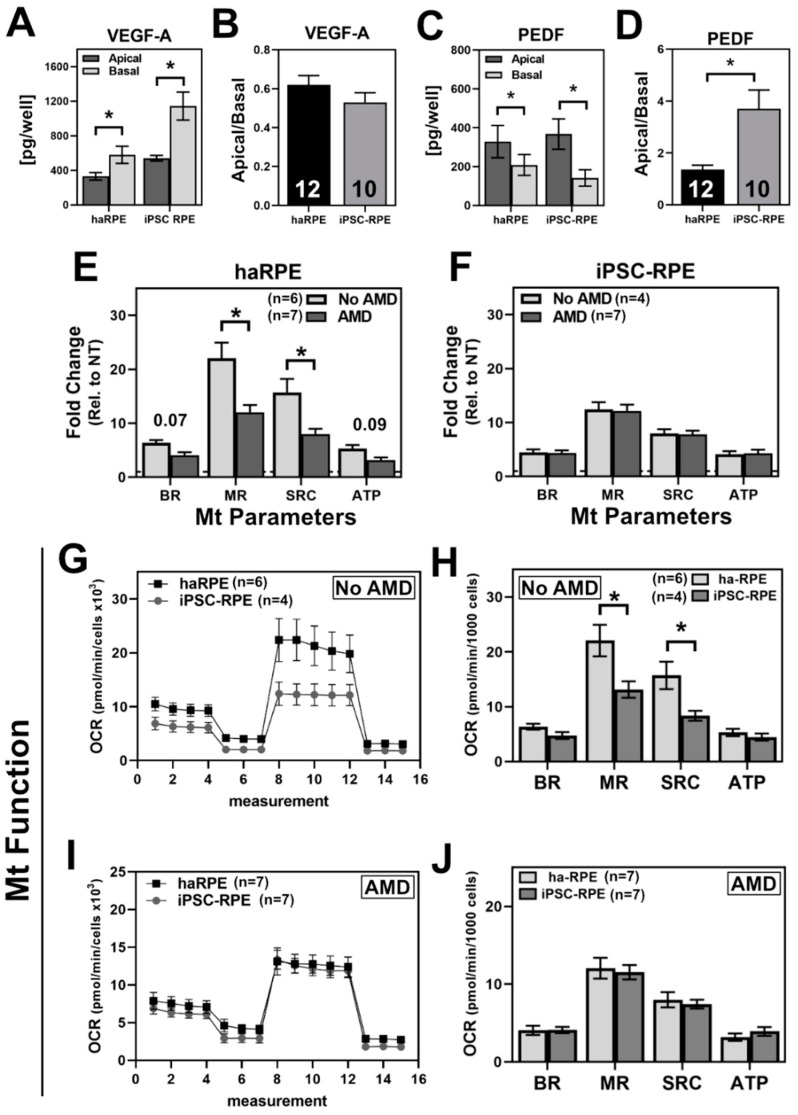
Basal characterization of haRPE and iPSC-RPE. (**A**,**C**) Quantification of VEGF-A (**A**) and PEDF (**C**) from apical and basal chambers of transwell plates for both haRPE and iPSC-RPE. (**B**,**D**) Ratio of apical/basal VEGF-A (**B**) and PEDF (**D**). (**E**,**F**) Mitochondrial (Mt) functional parameters calculated from OCR traces comparing disease states within haRPE (**E**) or iPSC-RPE (F). (**G**,**I**) Traces of the oxygen consumption rate (OCR) for haRPE and iPSC-RPE. (**H**,**J**) Mitochondrial functional parameters calculated from OCR traces. Sample size is indicated in each panel. * denotes *p* < 0.05.

**Figure 2 antioxidants-11-00605-f002:**
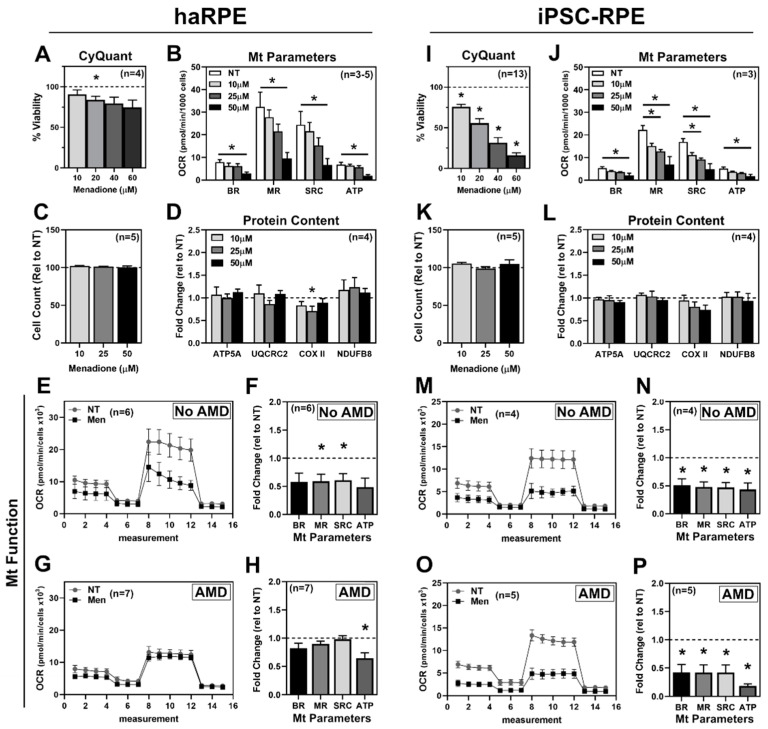
Response to the mitochondrial stressor menadione. haRPE (**A**–**H**) or iPSC-RPE (**I**–**P**) were treated with different doses of menadione for 24 h. (**A**,**I**) Cell viability measured under non-confluent conditions. (**B**,**J**) Mitochondrial (Mt) function as measured by CMST. (**C**,**K**) Cell count under confluent conditions. (**D**,**L**) Quantification of mitochondrial proteins. (**E**,**M**) OCR trace from No AMD donors for no treatment (NT) or after treating with 25 μM menadione. (**F**,**N**) Mitochondrial functional parameters for No AMD donors. (**G**,**O**) OCR trace from AMD donors for no treatment (NT) or after treating with 25 μM menadione. (**H**,**P**) Mitochondrial functional parameters from AMD donors. Sample size is indicated in each panel. * denotes *p* < 0.05.

**Figure 3 antioxidants-11-00605-f003:**
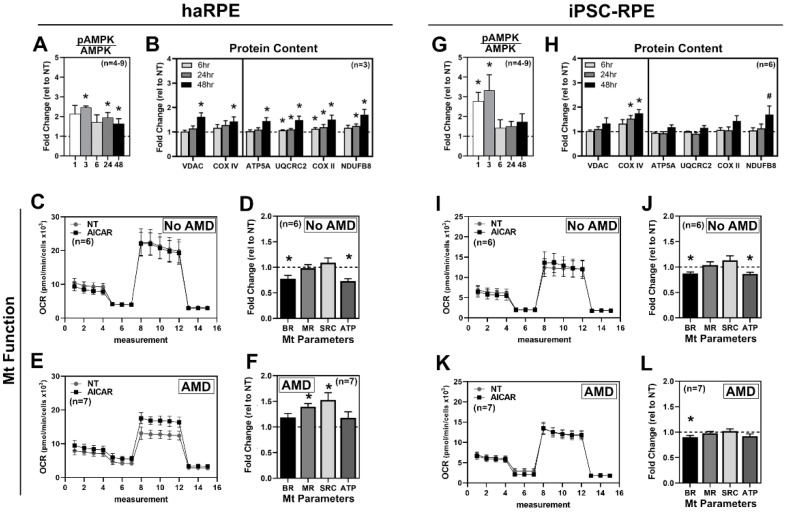
Response to AICAR. haRPE (**A**–**F**) or iPSC-RPE (**G**–**L**) were treated with 500 μM AICAR, then assayed at the indicated time. (**A**,**G**) pAMPK to AMPK ratio from Western immunoblots. (**B**,**H**) Quantification of mitochondrial proteins. (**C**,**I**) OCR trace from No AMD donors for no treatment (NT) or after treatment. (**D**,**J**) Mitochondrial (Mt) functional parameters from No AMD donors. (**E**,**K**) OCR trace from AMD donors for no treatment (NT) or after treatment. (**F**,**L**) Mitochondrial functional parameters from AMD donors. Sample size is indicated in each panel. * denotes *p* < 0.05.

**Figure 4 antioxidants-11-00605-f004:**
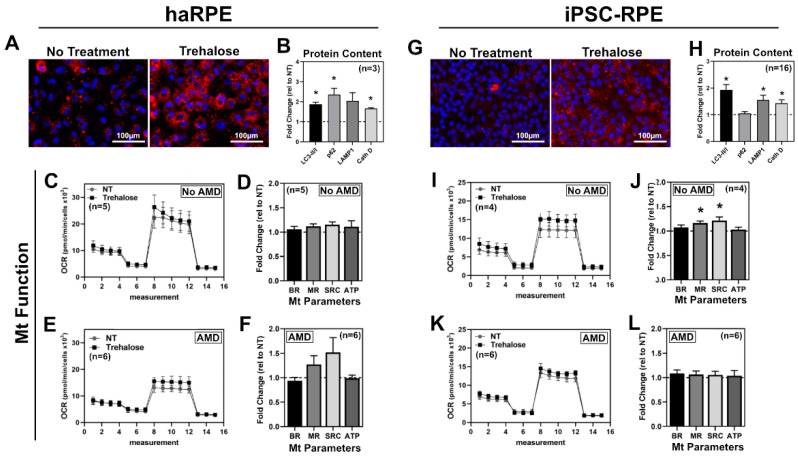
Response to trehalose. haRPE (**A**–**F**) or iPSC-RPE (**G**–**L**) were treated with 100 mM trehalose for 48 h. (**A**,**G**) Lysotracker (red) and nuclei (blue) max-intensity projections of 20× magnification images from no treatment (left) or after trehalose treatment (right). (**B**,**H**) Quantification of autophagy and lysosmal proteins. (**C**,**I**) OCR trace from No AMD donors for no treatment (NT) or after treatment. (**D**,**J**) Mitochondrial (Mt) functional parameters from No AMD donors. (**E**,**K**) OCR trace from AMD donors for no treatment (NT) or after treatment. (**F**,**L**) Mitochondrial functional parameters from AMD donors. Sample size indicated in panel for each assay. * denotes *p* < 0.05.

**Figure 5 antioxidants-11-00605-f005:**
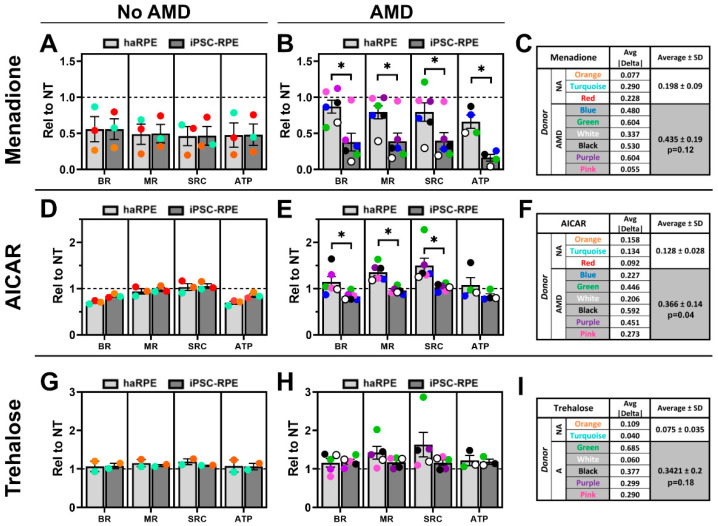
Comparison of haRPE and iPSC-RPE from the same donor. (**A**,**D**,**G**) Functional parameters calculated from OCR of paired cells from No AMD donors after treatment with menadione (**A**), AICAR (**D**), and trehalose (**G**). (**B**,**E**,**H**) Functional parameters calculated from OCR of paired cells from AMD donors after treatment with menadione (**B**), AICAR (**E**) and trehalose (**H**). Results were compared using a paired *t*-test. * denotes *p* < 0.05. (**C**,**F**,**I**) Summary of differences between haRPE and iPSC-RPE pairs across all four functional parameters. The average of the absolute value of the difference (|haRPE–iPSC-RPE|) for each parameter and the average for No AMD (NA) and AMD (**A**) donors is shown. Results of a paired t-test comparing No AMD to AMD is shown below AMD summary data.

**Table 1 antioxidants-11-00605-t001:** Demographics of donors—haRPE and iPSC-RPE matched pairs.

	Disease State ^A^	Age ^B^/ Gender ^C^	CFH ^D^/ARMS2 ^E^ Genotype	Cause of Death
Orange	No AMD	68/M	CT/GG	Cardiogenic shock
Turquoise	No AMD	84/F	CT/GG	Multi-system failure
Red	No AMD	73/M	TT/GG	Multiple myeloma
Blue	AMD	70/F	CC/GG	Sepsis
Green	AMD	75/F	CC/GT	Intracerebral bleed
White	AMD	66/M	TT/TT	Ischemic bowel
Black	AMD	83/F	CC/GG	Pancreatic cancer
Purple	AMD	75/F	CC/GG	Lung cancer
Pink	AMD	58/M	CT/GT	Acute cardiac event

^A^ Minnesota Grading System (MGS) was used to evaluate the stage of AMD in eye bank eyes [24]. No AMD = MGS1; AMD = MGS2 (early AMD) and MGS3 (intermediate AMD). ^B^ Age of donor, in years. ^C^ Gender of donor. F = female. M = male. ^D^ Complement Factor H (CFH) genotype for rs106117; low risk = TT, high risk = CT and CC. ^E^ Age-related maculopathy susceptibility 2 (ARMS2) for rs10490924; low risk = GG, high risk = GT and TT.

## Data Availability

Data used to support the findings in this study are contained within this article and the Appendix A.

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
