# Peer review of "Human iPSC- and Primary-Retinal Pigment Epithelial Cells for Modeling Age-Related Macular Degeneration"

_antioxidants, 2022, doi:10.3390/antiox11040605_

Round 1
Reviewer 1 Report
This manuscript describes functional differences between primary and derived RPE cells using a range of mitochondrial assays. The message is loud and clear: generally speaking cells from different sources may behave differently and therefore one need to be careful with interpreting results. There is always space for this kind of information, as there are a lot of papers that ignore potential confounders when interpreting data.
The manuscript extensively characterises the mitochondrial functions and show a range of similarities and differences between the RPE cells. I commend the authors for this well-conducted study.
There is one major shortfall of this paper. There is proper description of the condition the cells were cultured under and the time when these cells were used for the experiments. Reporting on TEER values and pigmentation is critical for experiments on RPE. If these were not assessed the secretion experiments has little value to decide whether the cells were in fact RPE. In short, it is imperative that the Authors include a full description of the seeding density, general conditions of culture, feeding regime, length in culture, pigmentation TEER under simultaneous conditions on culture inserts. It would also be highly desirable to show the morphology and general molecular characteristics of the used cells. The reason these are critical is because differences could simply come from different stages of differentiation. WHile the authors refer to previous publications, the fact that at some point an experiment work it did not have to work the next time. I am certain that information like these are readily available and can be incorporated into this manuscript.
A few minor issues:
1) abstract "human donors (haRPE)" is, in fact, human adult (haRPE) please define appropriately
2) line 430 inappropriate use of citation
3) line 393: why would higher passage help? Is there proof that these cells will "remember"? I think RPE is known to dedifferentiate at every passage.
4) discussion could be less speculative. While most points were raised I agree, the immediate question is why you the authors did not study meta genetic, metabolic memory etc?
Reviewer 2 Report
The authors have performed a comparative study between Human iPSC- and primary-retinal pigment epithelial cells as two experimental models for the study of age-related macular degeneration-related alterations in retinal pigment epithelium. The authors analyze cellular viability, growth factors secretion, and mitochondrial function in these two models derived from AMD and control patients. In addition, the cellular responses in terms of antioxidant defense, mitochondrial biogenesis, and autophagy were examined.The experiments are high quality and the results provide interesting information for the ophthalmology field. However, this reviewer has some questions, and minor comments and suggestions:
1- The information in table 1 and table S1 about the number of patients included is confusing. Why table 1 only includes 3 patients in the No-AMD group if in table S1 there are 12 for haRPE and 7 for iPSC-RPE? Also, If I have not misunderstood, haRPE and iPSC-RPE have been obtained from the same patient. Thus, why the sample size (n) is not the same in both cases?
On the other hand, the sample size is not indicated in all figure panels. For example, in figure 1E, where the p. values of some parameters of mitochondrial function are close to 0.05, it would be possible to suggest that by increasing the sample size, these differences could be statistically significant.
2- Authors should re-arrange the figure panels to make them easier to follow while reading the text. For example, in line 263 in the results section, fig. 3H appears before 3F (which is in line 271). Also, considering that the figures have a lot of panels, this reviewer strongly recommends adding explanation titles to the graphs.
3- The material and methods section should be more detailed. It should contain a complete explanation of how the experiments were performed, avoiding citations.
4- In lines 373-374, the authors discuss that reprogramming cells may remove the epigenetic signature of the adult somatic cell source, thereby reversing the “age” of iPSC-RPE. Have the authors compared the results between the younger patients with the older ones?
5- Finally, considering that women are more protected against oxidative stress due to the antioxidant effect of female hormones, have gender differences been found in any of the different measured parameters?
